# Discovery of New Cyclopentaquinoline Analogues as Multifunctional Agents for the Treatment of Alzheimer’s Disease

**DOI:** 10.3390/ijms20030498

**Published:** 2019-01-24

**Authors:** Kamila Czarnecka, Małgorzata Girek, Paweł Kręcisz, Robert Skibiński, Kamil Łątka, Jakub Jończyk, Marek Bajda, Jacek Kabziński, Ireneusz Majsterek, Piotr Szymczyk, Paweł Szymański

**Affiliations:** 1Department of Pharmaceutical Chemistry, Drug Analyses and Radiopharmacy, Faculty of Pharmacy, Medical University of Lodz, Muszyńskiego 1, 90d-151 Lodz, Poland; malgorzata.girek@stud.umed.lodz.pl (M.G.); pawel.krecisz@stud.umed.lodz.pl (P.K.); 2Department of Medicinal Chemistry, Faculty of Pharmacy, Medical University of Lublin, Jaczewskiego 4, 20-090 Lublin, Poland; robertskibinski@umlub.pl; 3Department of Physicochemical Drug Analysis, Chair of Pharmaceutical Chemistry, Faculty of Pharmacy, Jagiellonian University Medical College, Medyczna 9, 30-688 Krakow, Poland; kamil.latka@doctoral.uj.edu.pl (K.Ł.); jakub.jonczyk@doctoral.uj.edu.pl (J.J.); marek.bajda@uj.edu.pl (M.B.); 4Department of Clinical Chemistry and Biochemistry, Medical University of Lodz, Pl. Hallera 1, 90-647 Lodz, Poland; jacek.kabzinski@umed.lodz.pl (J.K.); ireneusz.majsterek@umed.lodz.pl (I.M.); 5Department of Pharmaceutical Biotechnology, Faculty of Pharmacy, Medical University of Lodz, Muszyńskiego 1, 90-151 Lodz, Poland; piotr.szymczyk@umed.lodz.pl

**Keywords:** acetylcholinesterase inhibitors, Alzheimer’s disease, molecular modeling, beta amyloid, yeast three-hybrid technology (Y3H) test

## Abstract

Here we report the two-step synthesis of 8 new cyclopentaquinoline derivatives as modifications of the tetrahydroacridine structure. Next, the biological assessment of each of them was performed. Based on the obtained results we identified 6-chloro-*N*-[2-(2,3-dihydro-*1H*-cyclopenta[b]quinolin-9-ylamino)-hexyl]]-nicotinamide hydrochloride (**3e**) as the most promising compound with inhibitory potencies against EeAChE and EqBuChE in the low nanomolar level 67 and 153 nM, respectively. Moreover, **3e** compound is non-hepatotoxic, able to inhibit amyloid beta aggregation, and shows a mix-type of cholinesterase’s inhibition. The mixed type of inhibition of the compound was confirmed by molecular modeling. Then, yeast three-hybrid (Y3H) technology was used to confirm the known ligand-receptor interactions. New derivatives do not show antioxidant activity (confirmed by the use of two different tests). A pKa assay method was developed to identify the basic physicochemical properties of **3e** compound. A LogP assay confirmed that **3e** compound fulfills Lipinsky’s rule of five

## 1. Introduction

Alzheimer’s disease (AD) is the most common form of dementia and chronic, neurodegenerative disease. AD attacks the brain leading to impaired memory, thinking and behavior, mainly among the elderly. AD has been proven to be a multifactorial disease associated with several aspects including low levels of acetylcholine (ACh), formation of β-amyloid (Aβ), hyperphosphorylated tau aggregates, oxidative stress and so on. Genetic studies have shown that dysfunction of Aβ or tau is sufficient to cause dementia. Ongoing molecular research is expected to lead to a true understanding of the disease’s pathogenesis. The neuropathological alterations described above suggest that the target at these factors could create the possible and effective treatment of AD [1,2].

The first and most frequently used therapeutic strategy was the cholinergic hypothesis. Nowadays, this approach is used as a treatment of mild to moderate stage AD. The acetylcholinesterase inhibitors (AChEIs) are the only ones that can produce significant therapeutic effects by enhancing the ACh level. So far, there have been only four AChEIs approved by the Food and Drug Administration (FDA) in the United States for the treatment of Alzheimer’s disease: tacrine (THA), donepezil, rivastigmine and galantamine (currently tacrine has been withdrawn). The last is memantine, a licensed AD drug, an uncompetitive *N*-Methyl-d-aspartate (NMDA) receptor antagonist. Unfortunately, none of the currently available strategies can completely prevent neurodegeneration and cure AD [2,3].

Current drug development for a treatment of Alzheimer’s disease is principally based on the amyloid cascade theory. The main goal is to reduce the levels of Aβ amyloid peptide in the brain. This can be achieved by decreasing peptide production through inhibition of β-secretase (also known as BACE-1) or by interfering with Aβ aggregation [4].

Tacrine (THA) is the first approved acetylcholinesterase (AChE) inhibitor for the treatment of Alzheimer’s disease and it has been extensively investigated in recent decades. Bis(7)-tacrine (B7T), which is a dimer of THA via a 7-carbon alkyl spacer, has shown much potent anti-AChE activity than THA [2].

One of the greatest challenges to the explanation of AD etiology is the difficulty in studying the earliest changes in neuronal function in the brain. The second challenge is to correlate these changes with ante mortem cognitive and behavioral function [5].

The failure to develop new pharmacotherapies of AD may be a result of focusing only on one targeting site of drug action. For this reason, research into new chemical compounds has been started on the basis of strategy “multi-target-directed ligands” (MTDLs) and a lot of clinical findings support this strategy for the treatment of AD. The chemical structure of AChEIs has been modified to achieve additional features, in order to solve the issue of both cholinergic deficiency and the other aforementioned targets in AD [6,7].

Currently, hundreds of novel AChEIs have been synthesized and investigated. The problem is only few of them investigated the properties relate to pharmaceutics such as the exact mechanism of action, cytotoxicity (safety of compounds), the penetration capacity across the blood–brain barrier into the brain or in vivo assays. Most studies end with knowing only the basic properties of the new compounds. 

Due to the constantly increasing number of people affected by AD and other forms of dementia, research into the causes and effective therapy is very important nowadays. Obtaining innovative compounds acting on the complex pathomechanism of this disease is very important and will provide valuable information for other scientists in the world. 

The ‘oxidative stress hypothesis’ of AD, of which reactive oxygen species (ROS) plays a key role in AD onset and progression, is well known. The relationship between oxidative stress and neurodegeneration is complex. The production of aberrant amounts of hydrogen peroxide is a general feature of aging and is linked with neurodegeneration [8,9].

In the presented research, we performed many of in silico and in vitro tests that allow us to precisely determine the mechanisms of action of designed and synthesized dimers. This stage of research is indispensable to select compounds for further in vivo studies. Thus, in this work we synthesized and evaluated new cyclopentaquinoline derivatives (**3a**–**3h**) (Scheme 1). The idea to modify tetrahydroacridine diamines into cyclopentaquinoline diamines was obtained from previously published research by Szymański et al. 2012 [10]. We decided to obtain new cyclopentaquinoline hybrids because the previously received cyclopentaquinoline derivatives with 5,6-dichloronicotinic acid gave very good results in the conducted research [11]. Among the tested compounds, we identified **3e** hybrid as a novel, non-hepatotoxic, not having antioxidant properties, AChE and BuChE inhibitor, and amyloid beta aggregation inhibitor, for potential AD therapy. We performed a pKa assay to identify the basic physicochemical properties of **3e** and logP to confirm the fulfillment of Lipinski’s rule of five for this compound, and perform the computer prediction of absorption, distribution, metabolism, excretion and toxicity (ADMET). Moreover, we conducted an innovative study: a yeast three-hybrid (Y3H) technology which is an extension of the two-hybrid (Y2H) system. Y3H technology was used to confirm the known ligand-receptor interactions. Contrary to the in vitro tests, the Y3H screens evaluate the putative receptor-ligand interactions in the in vivo conditions, making them more probably to occur in living organisms and strengthening their biological importance.

## 2. Results and Discussion

### 2.1. Chemistry

The synthetic route of novel cyclopentaquinoline hybrids **2a**–**2h** and **3a**–**3h** synthesized from eight diamines derivatives (**1a**–**1h**) has been reported in Scheme 1. The designed compounds were obtained using intermediates **1a**–**1h** prepared based on the previously described method [10]. In the first step, cyclopentaquinoline derivatives with proper diamines (**1a**–**1h**) reacted with 6-chloronicotinic acid to give compound **2a**–**2h**. Compounds **2a**–**2h** were obtained with satisfactory yield (70–84%) and purified by flash chromatography. Finally, compounds **3a**–**3h** were obtained by dissolving in an appropriate volume of methanol and hydrochloric acid with recrystallization from HCl in ether. All target compounds were characterized by ^1^ H NMR, ESI-MS, MS-HR and IR.

### 2.2. Biological Evaluation

#### 2.2.1. In Vitro Inhibition Studies on AChE and BuChE

All new cyclopentaquinoline derivatives (**3a**–**3h**) were evaluated in vitro for their inhibition of electric eel AChE (EeAChE) and equine serum BuChE (EqBuChE) using the modified Ellman assay [12,13]. The obtained IC_50_ values (50% inhibitory concentration) for compounds and reference are summarized in Table 1. The 4 of 8 compounds showed IC_50_ against EeAChE below 100 nM and the others below 600 nM. The compounds which are the most active inhibitors have longer alkyl linker, from 6 to 9 carbons in the chain. Tacrine was used as a positive control with an IC_50_ value of 81 and 20 nM for AChE and BuChE, respectively. Compared to tacrine, the **3e**–**3h** compounds showed higher inhibitory activity for AChE inhibition. Novel compounds showed satisfactory inhibition potency against BuChE with IC_50_ values ranging from 42 to 662 nM. Due to the highest inhibitory activity against AChE (IC_50_ = 67 nM), **3e** compound was chosen to the kinetic analysis of AChE inhibition and further research in order to the search for active compound in the treatment of Alzheimer’s disease.

#### 2.2.2. Kinetic Evaluation of Compound **3e**

To obtain the mechanism of EeAChE inhibition, kinetic experiments were performed. As shown in Figure 1, the interception of the lines in a Lineweaver–Burk plot occur above the x-axis into the same point. Plots and Km and Vmax values suggest that **3e** compound show typically mixed inhibition. Graphical analysis of Lineweaver–Burk plots of AChE activity showed increasing slopes and intercepts (decreased Vmax) with increasing inhibitor concentration.

#### 2.2.3. β-Amyloid Assay

Aβ(1–42) is the most amyloidogenic Aβ fragment found in the AD plaques. Besides the ability to inhibit AChE and BuChE, **3e** also inhibited in vitro self-induced Aβ (1–42) aggregation (Table 2). Compound **3e** was tested at four concentrations (10, 25, 50 and 100 µM) with Aβ (1–42) and was able to inhibit fibril formation by (79 ± 3.2)% in the highest concentration. But the effect was already significant at the lowest concentration (10 µM) (40 ± 4.8)% and the IC_50_ was 15 ± 0.6 µM.

#### 2.2.4. In Vitro Cytotoxicity Assay

Cytotoxicity assay was performed on CCL-110 cell line (normal, human fibroblasts derived from skin). Test over a wide range of concentrations (including the concentrations close to the IC_50_ of enzymatic inhibition results) was performed. In the range of concentrations 0.109–1.091 μM, the cells viability for **3e** was in the range of 41.90%–62.00%. Cells were incubated with a positive control–methanol, and had the viability of 22%. The obtained results for **3e** showed that the IC_50_ (0.638 μM) of the cytotoxic effect was 10 times lower than the IC_50_ (0.067 µM) of AChE inhibition. 

#### 2.2.5. In Vitro Hepatotoxicity

To investigate potential hepatoprotection activity and compare toxicity of **3e** and THA, human hepatic stellate cells (HSCs) were subjected to different concentrations of **3e** and THA. The range of concentration was chosen on the basis of IC_50_ results from AChE inhibition test. Cells viability was determined by MTT (3-(4, 5-dimethylthiazolyl-2)-2, 5-diphenyltetrazolium bromide) cytotoxicity test. For concentration 10 µM, higher viability of HSCs (lower hepatotoxicity) was shown for **3e** (93% ± 1.87) than for THA (90% ± 3.75). For concentrations 1 µM and 0.1 µM, viability for **3e** and THA was significant (99.65% ± 1.91, 100.18% ± 4.07 and 99.48% ± 0.74, 100.67% ± 1.03, respectively). In the one-way analysis of variance (ANOVA), results showed significant difference with *p* ≤ 0.05 among all doses used for **3e** and THA. Moreover, compound **3e** was much less toxic toward HSCs than CCL-110 cells. 

#### 2.2.6. In Vitro Inhibition Study on Hyaluronidase (HYAL)

Inflammation is a part of immune system and a body response to the pathogens or mechanical injury. Inflammation is controlled by cytokines, chemokines and several cellular enzymes (for example hyaluronidase). Hyaluronidase causes depolymerysation of hyaluronan, which is a part of extracellular matrix. Therefore, hyaluronidase weakens the integrity of tissues during inflammation. The inflammatory process is divided into two types—acute and chronic inflammation. Chronic inflammation is commonly associated with the development of many diseases such as cancer, rheumatoid arthritis or Alzheimer’s disease. Nowadays, non-steroidal anti-inflammatory drugs (NSAIDs) are in common use. Due to the many side effects such as gastrointestinal, renal or cardiovascular toxicity, these drugs use should be limited. [14,15] Therefore, a novel compound with an anti-inflammatory property was synthesized and tested.

The inhibitory activity of novel **3e** compound was tested by turbidimetric assay. [13] The values of new **3e** compound and a positive control–heparin were obtained. **3e** compound possesses small inhibitory activity towards hyaluronidase (IC_50_ 651 ± 1.48 µM), when positive control shows much higher inhibitory activity (IC_50_ 56 ± 0.78 µM). It can be concluded that the **3e** compound has minor anti-inflammatory property and slightly might help decrease inflammation. 

#### 2.2.7. LogP and pKa Assay

LogP is one of properties used in Lipinski’s rule of five. Our compounds were designed as a potential drug in AD therapy. Fulfillment of Lipinski’s rule of five allows us to suppose that the compound will show adequate pharmacokinetics. Compound **3e** shows the best activity, therefore we decided to obtain precise experimental physicochemical properties for these compounds, which are useful by estimating pharmacokinetics. In the chemical structure of our compound occur four nitrogen atoms with a free ion pair, which allow us to gradually ionize the whole compound at different pH value. Our procedure required a neutral form of compound, and therefore we performed simple and fast pKa assay to obtain necessary physicochemical properties of compound **3e** before logP assay. Direct determination of the ratio between consecutive form of molecule allows to measure the pKa value (Figure 2). The change of distribution charge in molecule affects noticeably the change of the ultraviolet (UV) spectra. The calculation on UV spectra was performed according to methodology developed by Musil et. al. Our method allows us to determine two pKa values of our compound in pH range 5.6 to 12.4. We used absorbance ratios and mathematical calculation allowed to estimate specific pH value for each ionized form and finally to calculate pKa values (Figure 3). The value pKa1 calculated by ChemAxon and ACD/Percepta software are close to value pKa1 obtained by our method. pKa2 value calculated by ChemAxon software and pKa2 calculated by ACD/Percepta were inflated in comparison with our experimental result pKa2 (Table 3).

Our procedure allowed fast, simple and cheap determination of the logP value of the most active compound **3e**. After pKa assay we decided to use TEA because of the best properties for the assay. Coefficient of determination was over 0.96 for calibration curve (Figure 3). Substances used to prepare calibration curve were listed in Table 4. The logP value for our test **3e** compound was 3.990. LogP values calculated by ChemAxon nad ACD/Percepta software were inflated in comparison to our experimental result (Table 3). The result confirms that the test compound fulfills Lipinski’s rule of five, and allows us to believe that compound **3e** can be useful in AD therapy.

#### 2.2.8. Yeast Three-Hybrid Technology (Y3H) Test

The positive and negative controls performed on the Matchmaker Y2H system gave expected results. Moreover, the bait (AChE) showed no autoactivation, enabling the experiment to proceed further. Initially the Y2H screen was performed to evaluate if the bait (AChE) interacts with any preys (i.e. human amyloid beta A4 protein (A4), human beta-secretase 1A (BACE-1A), human monoamine oxidase B (MAO B) and human microtubule associated protein tau (MAPT)) on the basis of protein–protein interactions that could interfere with planned Y3H procedure. Results of small-scale mating indicated no blue colonies on DDO/X/A (double dropout medium lacking tryptophan and leucine and supplemented with X-α-Gal and Aureobasidin A) agar plates. Negative results were observed for all four combinations of bait and each one of four analysed preys.

Obtained results encouraged to perform the Y3H test in the same combinations of bait and preys. Blue colonies, indicating putative interaction were observed on DDO/X/A agar plates for **3e** hybrid-ligand in the case of AChE-BACE-1A protein pair. The remaining three combinations of bait and preys (AChE-A4, AChE-MAO B, AChE-MAPT) gave negative results for hybrid ligands. Interactions remained stable on the more stringent QDO/X/A agar and diminished after removing the hybrid ligand. The results obtained support the view that three tested hybrid ligands induce interaction only between AChE and BACE-1A.

#### 2.2.9. ADMET (Absorption, Distribution, Metabolism, Excretion, Toxicity) Analysis

Our studies show that **3e** is the most active compound among our derivative. Because **3e** has the best properties, ADMET prediction was performed. We used experimental values of logP and pKa (base) obtained from the test described above. The most important property for potential medicine against Alzheimer’s disease (AD) is good central nervous system (CNS) penetration. **3e** presents a good blood–brain barrier permeation, logPS value was equal to −1.7. The compound can penetrate brain tissue, and logBB was equal 0.37 with fraction unbound in plasma 0.062 and fraction unbound in brain 0.03. The probability of positive Ames test was 0.35 which means a potential low genotoxicity effect. Our compound fulfills Lipinski’s “Rule of five”. Hydrogen bond donors is lower than 5, hydrogen bind acceptors is lower than 10, molecular weight is lower than 500, logP is lower than 5, and TPSA is lower than 140. To sum up, compound **3e** has a good profile as a potential AD drug.

#### 2.2.10. Antioxidant Activity Assay

Reactive oxygen species are a causal factor in the process of ageing and cumulative damage has been associated with AD. [9] The antioxidant activities of target compounds **3a**–**3h** were evaluated using a radical scavenging assay (DPPH assay) and ABTS assay with trolox as reference compound (Table 1). Connecting the 6-chloronicotinic acid with cyclopentaquinoline moiety with variation of linker length between the two fragments lower the values of the antioxidant activity compared to trolox. A significant amount does not have antioxidant properties confirmed by two assays. In the ABTS assay, the best antioxidant activity have **3b** compound with FRS_50_ 1.9 ± 0.30 but it is 48 times less active than the reference compound. In the DPPH assay, none of the new derivatives showed antioxidant activity and the **3e** compound do not exert radical scavenging properties.

#### 2.2.11. Neuroprotection Against Oxidative Stress

Neuroprotective property of novel synthesized compound **3e** against oxidative stress was determined on SH-SY5Ycells in three experiments. **3e** was tested in the range of low concentrations, according to IC_50_ results of AChE and BuChE inhibitions assays. The first model regarded generation of exogenous free radicals by H_2_O_2_. Cells were incubated with **3e** for 24 h before the addition of toxic stimulus. After incubation, H_2_O_2_ (100 µM) was added and the cells were maintained for 24 h in the presence of the compounds. **3e** and reference compound, trolox, were tested in four concentration—10, 1, 0.1 µM and 0.01 µM. Control cells without H_2_O_2_ and the compound had viability of 100%. Positive control cells incubated only with H_2_O_2_ (without **3e**) had a viability of 84.27%. Cells treated with trolox at the concentrations of 10, 1, 0.1, 0.01 µM had viability of 99.22%, 99.38%, 98.62% and 96.41%, respectively, and neuroprotection of 95.05%, 96.03%, 91.22% and 77.16%, respectively. Unfortunately, at none concentration, **3e** had neuroprotective activity. Its viability was of 14.34%, 18.42%, 29.24% and 60.39% at concentrations of 10, 1, 0.1, 0.01 µM, respectively (Table 5). Incubation of **3e** with toxic stimulus could result in strong increase of cytotoxicity, therefore neuroprotection wasn’t observed even at the lowest dose of **3e**. In the one-way ANOVA, values of *p* ≤ 0.05 were considered statistically significant for **3e**, not for trolox. A R/O (Rotenone/Oligomycin A) mixture was used to induce mitochondrial ROS by blocking mitochondrial electron transport chain (complexes I and V). [21] In the pre-incubation study it can be checked if compound has neuroprotective property due to the activation of endogenous antioxidants pathways. In the co-incubation it can be determined if a compound is a free-radical scavenger [22]. The MTT test was used because of the measuring of mitochondrial activity. Apoptotic or necrotic cells do not make this chemical modification, whereas live cells reduce MTT. In the first experiment, SH-SY5Y cells were incubated with **3e** for 24 h before the addition of toxic stimulus. Next, R/O mixture was added, and cells were maintained for 24 h in the presence of **3e**. In the pre-incubation assay, SH-SY5Y cells were treated with **3e** in the range of concentration 0.1–0.0001 µM (Table 1). Trolox was used as a reference compound. Control cells without R/O mixture and compound had a viability of 100%. Positive control cells exposed to the R/O mixture showed viability of 46.77% in the incubation without the presence of **3e**. Compound **3e** did not have a neuroprotective property, except from concentration of 0.0001 µM. At this concentration, neuroprotection was of 3.98%. Trolox showed neuroprotection at the concentration of 0.001 µM with a value of 2.15%. One-way ANOVA was performed and results of **3e** were statistically significant (*p* ≤ 0.05). In the co-incubation experiment, **3e** and R/O mixture were incubated together for 24 h. Cells exposed to the R/O mixture showed viability of 47.91% in the incubation without the presence of **3e**. Trolox had small neuroprotection at the concentrations of 0.01 µM and 0.001 µM, 4.67% and 5.60%, respectively. Trolox has stronger neuroprotection properties in higher concentration. In the one-way ANOVA, the results were not statistically significant (*p* ≤ 0.05) [23]. **3e** did not have neuroprotective properties at any concentrations and was not able to capture generated radicals (Table 5).

#### 2.2.12. Molecular Modeling

The binding mode of obtained compounds with AChE and BuChE was studied by docking to the enzyme active sites. While docking to AChE, for all compounds the fragment of tacrine analogue with a cyclopentane ring was located in anionic site. This created a characteristic sandwich due to π–π stacking and cation–π interaction with Trp84 and Phe330. A protonated nitrogen atom was engaged in the hydrogen bond with the carbonyl group of the main chain of His440. Significant differences in arrangement were observed for the 6-chloronicotinamide fragment. In case of compounds with short and medium length carbon linker (**3a**, **3b**, **3c**, **3d**) poses were in bent conformation, in which the 6-chloronicotinamide fragment formed hydrophobic interactions mainly with Tyr334, Phe331, Phe330, Phe290 and Tyr121. The most active compound **3e**, presented consistently high-rated poses in extended conformation, in which chloropyridine ring created π–π stacking with Trp279 and Tyr70 as well as CH–π interaction with Tyr121 in the peripheral anionic site. Additionally, an amide nitrogen atom formed a hydrogen bond with the hydroxyl group of Tyr121 (Figure 4A). This binding mode explained the high activity of this compound. Derivatives with long carbon linkers (**3f**, **3g**, **3g**) and slightly lower activity toward AChE, were located similarly to compound **3e**; however, creation of a hydrogen bond with Tyr121 was difficult (Figure 5A).

While docking to BuChE, the arrangement of cyclopentaquinoline moiety was very similar to that observed for AChE. This created π–π stacking with Trp82, and a hydrogen bond with the main chain of His438. In case of compounds with a short carbon linker (**3a**, **3b**), 6-chloronicotinamide moiety occupied the hydrophobic pocket created by Phe329, Phe398 and Trp231. In this position, CH–π interaction with Phe329 and interaction between the chlorine atom and indole ring of Trp231 were observed. For compound **3a**, poses were more consistent and were reflected in the higher activity. Compounds **3c** and **3d** also were in a bent conformation. However, previously described interactions with Phe329 and Trp231 were difficult to obtain due to the longer linker. Compounds with a long carbon linker (**3e**–**3h**) occurred in extended conformation (Figure 4B and Figure 5B). Among them, compound **3g** with the highest activity toward BuChE, presented the most consistent poses. Its 6-chloronicotinamide fragment created hydrophobic interactions with Pro285, Tyr282 and Ile356 near the entrance to the active site, as well as a hydrogen bond with the main chain of Tyr332 in the peripheral anionic site (Figure 5B). 

## 3. Materials and Methods

### 3.1. Synthesis

All the analytical grade reagents were purchased from SigmaAldrich. The chemical reactions were monitored by thin-layer chromatography (TLC). The solvents were removed by rotary evaporation under reduced pressure. Flash chromatography as the purifying technique was performed using silica gel 60, Merck. infrared (IR) spectra were recorded on the Mattson Infinity Series Fourier transform infrared (FT-IR) spectrophotometer, in ATR. ^1^H NMR spectra were recorded on BrukerAvance III 600 MHz spectrometer. Tetramethylsilane was used as the internal standard. Mass spectra were acquired using the Agilent Accurate Mass Q-TOF LC/MS G6520B as dual electrospray source and Infinity 1290 ultrahigh-pressure system with performance liquid chromatography comprising of: a binary pump G4220A, FC/ALS thermostat G1330B autosampler G4226A DAD detector G4212A and TCC G1316C module (Agilent Technologies, Santa Clara, CA, USA). In a positive mode Q-TOF detector was tuned with use of Agilent ESI-L tuning mix of high resolution mode (4 GHz). Melting points were measured using the electrothermal apparatus. 

#### 3.1.1. General Procedure for the Synthesis of Compound **1a**–**1h**

Compounds **1a**–**1h** were synthesized according to the previously described method [10,11]. To the flask of tetrahydrofuran (THF) (10 mL), 2-chloro-4,6-dimethoxy-1,3,5-triazine (CDMT), 6-chloronicotinic acid and *N*-methylomorpholine were added. The reaction was carried out for about 2 h in an ice bath until the CDMT was consumed. Next to the mixture, compounds **1a**–**1h** dissolved in THF (3 mL) were added and then stirred for 24 h at room temperature.

#### 3.1.2. General Procedure for the Synthesis of Compounds **2a**–**2h**


A mixture of tetrahydrofuran (THF) (8 mL), 2-chloro-4,6-dimethoxy-1,3,5-triazine (CDMT) (0.27–0.39 g, 1.54–2.20 mM), 6-chloronicotinic acid (0.24–0.35 g, 1.54–2.20 mM) and dropwise of *N*-methylomorpholine (0.17–0.24 mL, 1.54–2.20 mM) were added. The reaction was carried out for about 2 h in an ice bath. Then (0.50 g, 1.54–2.20 mM) of proper diamine dissolved in THF (3 mL) was added to the mixture and the reaction continued at room temperature for 24 h with stirring. Purification was performed by flash chromatography to give compounds **2a**–**2h**. Physical and spectral data are listed below.

##### 6-chloro-*N*-[2-(2,3-dihydro-1H-cyclopenta[b]quinolin-9-ylamino)ethyl]pyridine-3-carboxamide (**2a**)

Compound **2a**: beige solid (74% yield): mp 77–80 °C; IR (KBr)*v* (cm^−1^): 1659.3, 2934.3, 3029.8, 3243.0;^1^H NMR (600 MHz, Methanol-*d_4_*)(δ ppm.): 8.75 (1H, s, Ar), 8.36 (1H, d, *J* = 8.5 Hz, Ar), 8.19 (1H, d, *J* = 10.9 Hz, Ar), 7.88 (1H, t, *J* = 7.7 Hz, Ar), 7.77 (1H, d, *J* = 8.3 Hz, Ar), 7.68 (1H, t, *J* = 8.3 Hz, Ar), 7.57 (1H, d, *J* = 8.4 Hz, Ar), 4.12 (2H, t, *J* = 6.0 Hz), 3.79–3.81 (2H, m, CH_2_), 3.44–3.49 (2H, m), 3.20 (2H, t, *J* = 7.9 Hz), 2.32 (2H, p, *J* = 7.7 Hz, CH_2_), (protons of NH groups invisible); MS (ESI)(M+1) *m/z*: 367.1, 185.1, 140.0, 121.1; MS-HR (ESI): calcd. for C_20_H_19_ClN_4_O: 366.12474, found: 366.12450.

##### 6-chloro-*N*-[3-(2,3-dihydro-1H-cyclopenta[b]quinolin-9-ylamino)propyl]pyridine-3-carboxamide (**2b**)

Compound **2b**: beige solid (80% yield): mp 67–70 °C; IR (KBr)*v* (cm^−1^): 1668.5, 2936.0, 3017.6, 3291,0;^1^H NMR (600 MHz, Methanol-*d*_4_) (δ ppm.): 8.80 (1H, s, Ar), 8.41 (1H, d, *J* = 8.5 Hz, Ar), 8.22 (1H, d, *J* = 8.3 Hz, Ar), 7.87 (1H, t, *J* = 7.2 Hz, Ar), 7.76 (1H, d, *J* = 8.4 Hz, Ar), 7.66 (1H, t, *J* = 7.8 Hz, Ar), 7.57 (1H, d, *J* = 8.3 Hz, Ar), 3.90–3.97 (2H, m, CH_2_), 3.61 (2H, t, *J* = 6.5 Hz, CH_2_), 3.41 (2H, t, *J* = 7.3 Hz, CH_2_), 3.21 (2H, q, *J* = 7.5 Hz, CH_2_), 2.30 (2H, p, *J* = 7.8 Hz, CH_2_), 2.09 (2H, p, *J* = 6.7 Hz, CH_2_), (protons of NH groups invisible); MS (ESI)(M+1) *m/z*: 381.1, 185.1, 169.0, 140.0, 121.1; MS-HR (ESI) (M+1): calcd. for C_21_H_21_ClN_4_O: 380.14039, found: 380.14040.

##### 6-chloro-*N*-[3-(2,3-dihydro-1*H*-cyclopenta[*b*]quinolin-9-ylamino)butyl]pyridine-3-carboxamide (**2c**)

Compound **2c**: beige solid (72% yield): mp 62–65 °C; IR (KBr) *v* (cm^−1^): 1642.0, 2933.2, 3036.1, 3253.0;^1^H NMR (600 MHz, Methanol-*d*_4_) (δ ppm.): 8.76 (1H, s, Ar), 8.35 (1H, d, *J* = 8.6 Hz, Ar), 8.19 (1H, d, *J* = 2.5 Hz, Ar), 7.87 (1H, t, *J* = 7.7 Hz, Ar), 7.75 (1H, d, *J* = 8.0 Hz, Ar), 7.65 (1H, t, *J* = 7.8 Hz, Ar), 7.56 (1H, d, *J* = 7.9 Hz, Ar), 3.87–3.92 (2H, m, CH_2_), 3.49 (2H, t, *J* = 6.8 Hz, CH_2_), 3.40–3.43 (2H, m, CH_2_), 3.20 (2H, q, *J* = 9.2, 8.0 Hz, CH_2_), 2.27-3.32 (2H, m, CH_2_), 1.79–1.91 (4H, m, CH_2_), (protons of NH groups invisible); MS (ESI)(M+1) *m/z*: 395.2, 239.2, 185.1, 140.0, 121.1; MS-HR (ESI) (M+1): calcd. for C_22_H_23_ClN_4_O: 394.15604, found: 394.15782.

##### 6-chloro-*N*-[3-(2,3-dihydro-1*H*-cyclopenta[*b*]quinolin-9-ylamino)pentyl]pyridine-3-carboxamide (**2d**)

Compound **2d**: beige solid (84% yield): mp 101–103 °C; IR (ATR)*v* (cm^−1^): 761.2; 1242.6; 1363.3; 1447.9; 1561.5; 2937.9; 3234.5;^1^H NMR (600 MHz, Methanol-*d*_4_) (δ ppm.): 8.69 (1H, s, Ar), 8.31 (1H, d, *J* = 8.3 Hz, Ar), 8.11 (1H, d, *J* = 8.3 Hz, Ar), 7.85 (1H, t, *J* = 7.9 Hz, Ar), 7.72 (1H, d, *J* = 8.4 Hz, Ar), 7.61 (1H, t, *J* = 7.8 Hz, Ar), 7.52 (1H, d, *J* = 8.4 Hz, Ar), 3.83–3.88 (2H, m, CH_2_), 3.43–3.46 (2H, m, CH_2_), 3.39–3.42 (2H, m, CH_2_), 3.16–3.21 (2H, m, CH_2_), 2.27–2.34 (2H, m, CH_2_), 1.81–1.88 (2H, m, CH_2_), 1.74 (2H, p, *J* = 7.4 Hz, CH_2_), 1.53–1.59 (2H, m, CH_2_), (protons of NH groups invisible); MS (ESI)(M+1) *m/z*: 409.2, 253.2, 185.1, 140.0; MS-HR (ESI): calcd. for C_23_H_25_ClN_4_O: 408.17169, found: 408.17200.

##### 6-chloro-*N*-[3-(2,3-dihydro-1*H*-cyclopenta[*b*]quinolin-9-ylamino)hexyl]pyridine-3-carboxamide (**2e**)

Compound **2e**: beige solid (76% yield): mp 123–125 °C; IR (ATR)*v* (cm^−1^): 761.0; 1242.6; 1364.4; 1448.4; 1560.8; 2939.5; 3246.3; ^1^H NMR (600 MHz, Chloroform-*d*) (δ ppm.): 9.03 (1H, s, Ar), 8.32 (1H, d, *J* = 6.3 Hz, Ar), 8.12 (1H, d, *J* = 8.3 Hz, Ar), 7.85 (1H, t, *J* = 7.3 Hz, Ar), 7.67–7.70 (1H, m, Ar), 7.55–7.58 (1H, m, Ar), 7.38 (1H, d, *J* = 8.0 Hz, Ar), 3.77–3.82 (2H, m, CH_2_), 3.29 (2H, t, *J* = 7.7 Hz, CH_2_), 3.17–3.21 (2H, m, CH_2_), 2.90–2.95 (2H, m, CH_2_), 2.34–2.39 (2H, m, CH_2_), 2.17–2.27 (4H, m, CH_2_), 1.78–1.84 (2H, m, CH_2_), 1.72–1.76 (2H, m, CH_2_), (protons of NH groups invisible); MS (ESI)(M+1) *m/z*: 423.2, 267.2, 185.1, 140.0; MS-HR (ESI): calcd. for C_24_H_27_ClN_4_O: 422.18734, found: 422.18740.

##### 6-chloro-*N*-[3-(2,3-dihydro-1*H*-cyclopenta[*b*]quinolin-9-ylamino)heptyl]pyridine-3-carboxamide (**2f**)

Compound **2f**: beige solid (70% yield): mp 88–90 °C; IR (ATR)*v* (cm^−1^): 760.8; 1242.6; 1365.9; 1455.7; 1559.1; 2930.8; 3205.2; ^1^H NMR (600 MHz, Methanol-d_4_) (δ ppm.): 8.77 (1H, s, Ar), 8.33 (1H, d, *J* = 8.6 Hz, Ar), 8.19 (1H, d, *J* = 8.3 Hz, Ar), 7.87 (1H, t, *J* = 7.1 Hz, Ar), 7.75 (1H, d, *J* = 7.8 Hz, Ar), 7.63–7.66 (1H, m, Ar), 7.56 (1H, d, *J* = 8.9 Hz, Ar), 3.81–3.86 (2H, m, CH_2_), 3.41 (2H, t, *J* = 7.1 Hz, CH_2_), 3.19-3.23 (4H, m, CH_2_), 2.32 (2H, p, *J* = 7.8 Hz, CH_2_), 1.80 (2H, p, *J* = 7.6 Hz, CH_2_), 1.66 (2H, p, *J* = 7.3 Hz, 7H), 1.48 (6H, m, CH_2_), (protons of NH groups invisible); MS (ESI)(M+1) *m/z*: 437.2, 281.2, 185.1, 140.0; MS-HR (ESI): calcd. for C_25_H_29_ClN_4_O: 436.20299, found: 436.20374.

##### 6-chloro-*N*-[3-(2,3-dihydro-1*H*-cyclopenta[*b*]quinolin-9-ylamino)octyl]pyridine-3-carboxamide (**2g**)

Compound **2g**: beige solid (82% yield): mp 84–86 °C; IR (ATR)*v* (cm^−1^): 759.9; 1243.6; 1363.5; 1453.9; 1559.4; 2930.2; 3219.4; ^1^H NMR (600 MHz, Methanol-*d*_4_) (δ ppm.): 8.79 (1H, s, Ar), 8.36 (1H, d, *J* = 8.3 Hz, Ar), 8.20 (1H, d, *J* = 8.3 Hz, Ar), 7.86 (1H, t, *J* = 7.7 Hz, Ar), 7.77 (1H, d, *J* = 8.4 Hz, Ar), 7.64 (1H, t, *J* = 7.8 Hz, Ar), 7.56 (1H, d, *J* = 8.8 Hz, Ar), 3.83 (2H, q, *J* = 7.3 Hz, CH_2_), 3.40 (2H, t, *J* = 7.2 Hz, CH_2_), 3.19–3.22 (2H, m, CH_2_), 2.93–2.97 (2H, m, CH_2_), 2.28–2.34 (2H, m, CH_2_), 1.79 (2H, p, *J* = 7.7 Hz, CH_2_), 1.63–1.71 (4H, m, CH_2_), 1.47–1.53 (6H, m, CH_2_), (protons of NH groups invisible); MS (ESI) (M+1)*m/z*: 451.2, 295.2, 185.1, 140.0; MS-HR (ESI): calcd for C_26_H_31_ClN_4_O: 450.21864, found: 450.22025.

##### 6-chloro-*N*-[3-(2,3-dihydro-1*H*-cyclopenta[*b*]quinolin-9-ylamino)nonyl]pyridine-3-carboxamide (**2h**)

Compound **2h**: beige solid (80% yield): mp 109–111 °C; IR (ATR)*v* (cm^−1^): 764.4; 1242.0; 1364.8; 1457.1; 1560.4; 2925.5; 3190.2; ^1^H NMR (600 MHz, Methanol-*d*_4_) (δ ppm.): 8.79 (1H, s, Ar), 8.34 (1H, d, *J* = 8.4 Hz, Ar), 8.20 (1H, d, *J* = 8.3 Hz, Ar), 7.87 (1H,t, *J* = 8.2 Hz, Ar), 7.76 (1H, d, *J* = 8.4 Hz, Ar), 7.65 (1H, t, *J* = 7.2 Hz, Ar), 7.57 (1H, d, *J* = 8.3 Hz, Ar), 3.80–3.84 (2H, m, CH_2_), 3.36–3.43 (2H, m, CH_2_), 3.27 (2H, t, *J* = 9.9 Hz, CH_2_), 3.21 (2H, q, *J* = 8.0, 7.3 Hz, CH_2_), 2.32 (2H, p, *J* = 7.8 Hz, CH_2_), 1.79 (2H, p, *J* = 7.4 Hz, CH_2_), 1.63–1.70 (4H, m, CH_2_), 1.46-1.51 (8H, m, CH_2_), (protons of NH groups invisible); MS (ESI) (M+1) *m/z*: 465.2, 309.2, 185.1, 140.0; MS-HR (ESI): calcd. for C_27_H_33_ClN_4_O: 464.23429, found: 464.23570.

### 3.2. General Procedure for the Synthesis of Compounds ***3a***–***3h***

The compounds **2a**–**2h** (0.020 g) were dissolved in methanol (1 mL). Next HCl/ether (10 mL) was added. After 24 h precipitate had formed. Then the precipitate was isolated by filtration and dried, isolated by filtration, and dried. In this synthesis, the **3a**–**3h** compounds were obtained. Physical and spectral data are listed below.

#### 3.2.1. 6-chloro-*N*-[2-(2,3-dihydro-1*H*-cyclopenta[*b*]quinolin-9-ylamino)ethyl]pyridine-3-carboxamide hydrochloride (**3a**)

Yield: 56%; brown solid; mp 222–224 °C; IR (KBr)*v* (cm^−1^): 1657.6, 2936.8, 3242.1; ^1^H NMR (600 MHz, DMSO-*d*_6_) (δ ppm.): 14.13 (1H, s, HCl), 8.81 (1H, s, Ar), 8.53 (1H, d, *J* = 8.6 Hz, Ar), 8.24 (1H, d, *J* = 8.3 Hz, Ar), 7.83–7.89 (2H, m, Ar), 7.61-7.66 (2H, m, Ar), 3.96 (2H, q, *J* = 6.2 Hz, CH_2_), 3.61 (2H, q, *J* = 6.0 Hz, CH_2_), 3.26–3.30 (2H, m, CH_2_), 3.14 (2H, t, *J* = 7.9 Hz, CH_2_), 2.17 (2H, p, *J* = 7.6 Hz, CH_2_), (protons of NH groups invisible); MS (ESI) (M+1)*m/z*: 367.1, 185.1, 140.0, 121.1; MS-HR (ESI) calcd. for C_20_H_20_Cl_2_N_4_O: 366.12474, found: 366.12728.

#### 3.2.2. 6-chloro-*N*-[3-(2,3-dihydro-1*H*-cyclopenta[*b*]quinolin-9-ylamino)propyl]pyridine-3-carboxamide hydrochloride (**3b**)

Yield: 48%; brown solid; mp 168–170 °C; IR (KBr)*v* (cm^−1^): 1671.5, 2924.9, 3285.2; ^1^H NMR (600 MHz, DMSO-*d*_6_) (δ ppm.): 14.03 (1H, s, HCl), 8.84 (1H, s, Ar), 8.54 (1H, d, *J* = 8.5 Hz, Ar), 8.25 (1H, d, *J* = 8.3 Hz, Ar), 7.81–7.89 (2H, m, Ar), 7.59–7.65 (2H, m, Ar), 3.82 (2H, q, *J* = 6.7 Hz, CH_2_), 3.42–3.48 (2H, m, CH_2_), 3.24–3.31 (2H, m, CH_2_), 3.13 (2H, t, *J* = 7.9 Hz, CH_2_), 2.13 (2H, p, *J* = 7.8 Hz, CH_2_), 1.96 (2H, p, *J* = 6.7 Hz, CH_2_), (protons of NH groups invisible); MS (ESI) (M+1) *m/z*: 381.1, 185.1, 140.0, 121.1; MS-HR (ESI) calcd for C_21_H_22_Cl_2_N_4_O: 380.14039, found: 380.13994.

#### 3.2.3. 6-chloro-*N*-[3-(2,3-dihydro-1*H*-cyclopenta[*b*]quinolin-9-ylamino)butyl]pyridine-3-carboxamide hydrochloride (**3c**)

Yield: 54%; brown solid; mp 115–117 °C; IR (KBr)*v* (cm^−1^):1696.9, 2932.9, 3223.1; ^1^H NMR (600 MHz, DMSO-*d*_6_) (δ ppm.): 14.10 (1H, s, HCl), 8.82 (1H, s, Ar), 8.51 (1H, d, *J* = 8.6 Hz, Ar), 8.23 (1H, d, *J* = 8.3 Hz, Ar), 7.82–7.88 (2H, m, Ar), 7.6–7.64 (2H, m, Ar), 3.76 (2H, q, *J* = 6.8 Hz, CH_2_), 3.31–3.36 (2H, m, CH_2_), 3.27–3.31 (2H, m, CH_2_), 3.13 (2H, t, *J* = 7.9 Hz, CH_2_), 2.14 (2H, p, *J* = 7.8 Hz, CH_2_), 1.72–1.77 (2H, m, CH_2_), 1.64–1.69 (2H, m, CH_2_), (protons of NH groups invisible); MS (ESI) (M+1)*m/z*: 395.2, 298.2, 239.2, 185.1, 140.0, 121.1; MS-HR (ESI) calcd. for C_22_H_24_Cl_2_N_4_O: 394.15604, found: 394.15764.

#### 3.2.4. 6-chloro-*N*-[3-(2,3-dihydro-1*H*-cyclopenta[*b*]quinolin-9-ylamino)pentyl]pyridine-3-carboxamide hydrochloride (**3d**)

Yield: 63%; brown solid; mp 85–87 °C; IR (ATR)*v* (cm^−1^): 760.1; 1244.0; 1399.0; 1455.9; 1558.0; 2878.5; 3205.5; ^1^H NMR (600 MHz, DMSO-*d*_6_) (δ ppm.): 13.94 (1H, s, HCl), 8.79 (1H, s, Ar), 8.48 (1H, d, *J* = 8.7 Hz, Ar), 8.19 (1H, d, *J* = 8.3 Hz, Ar), 7.86 (1H, t, *J* = 7.7 Hz, Ar), 7.81 (1H, d, *J* = 7.7 Hz, Ar), 7.62 (1H, t, *J* = 7.8 Hz, Ar), 7.59 (1H, d, *J* = 6.7 Hz, Ar), 3.73 (2H, q, *J* = 6.9 Hz, CH_2_), 3.26–3.30 (4H, m, CH_2_), 3.14 (2H, t, *J* = 7.8 Hz, CH_2_), 2.15–2.20 (2H, m, CH_2_), 1.70–1.75 (2H, m, CH_2_), 1.61 (2H, p, *J* = 6.9 Hz, CH_2_), 1.44 (2H, p, *J* = 7.6, 6.9 Hz, CH_2_), (protons of NH groups invisible); MS (ESI) (M+1) *m/z*: 409.2, 253.2, 185.1, 140.0; MS-HR (ESI) calcd. for C_23_H_26_Cl_2_N_4_O: 408.17169, found: 408.17180.

#### 3.2.5. 6-chloro-*N*-[3-(2,3-dihydro-1*H*-cyclopenta[*b*]quinolin-9-ylamino)hexyl]pyridine-3-carboxamide hydrochloride (**3e**)

Yield: 48%; brown solid; mp 83–84 °C; IR (ATR) (cm^−1^): 758.6; 1275.1; 1360.0; 1456.2; 1557.2; 2936.3; 3249.9; ^1^H NMR (600 MHz, DMSO-*d*_6_) (δ ppm.): 13.96 (1H, s, HCl), 8.82 (1H, s, Ar), 8.49 (1H, d, *J* = 8.6 Hz, Ar), 8.23 (1H, d, *J* = 8.3 Hz, Ar), 7.85–7.89 (2H, m, Ar), 7.61–7.67 (2H, m, Ar), 3.72 (2H, q, *J* = 6.7 Hz, CH_2_), 3.25–3.30 (4H, m, CH_2_), 3.15 (2H, t, *J* = 7.9 Hz, CH_2_), 2.17 (2H, p, *J* = 7.8 Hz, CH_2_), 1.69 (2H, p, *J* = 7.7, 7.3 Hz, CH_2_), 1.56 (2H, p, *J* = 7.2 Hz, CH_2_), 1.35–1.46 (4H, m, CH_2_), (protons of NH groups invisible); MS (ESI) (M+1) *m/z*: 423.2, 267.2, 185.1, 140.0; MS-HR (ESI) calcd. for C_24_H_28_Cl_2_N_4_O: 422.18734, found: 422.18855.

#### 3.2.6. 6-chloro-*N*-[3-(2,3-dihydro-1*H*-cyclopenta[*b*]quinolin-9-ylamino)heptyl]pyridine-3-carboxamide hydrochloride (**3f**)

Yield: 50%; brown solid; mp 80–82 °C; IR (ATR)*v* (cm^−1^): 760.7; 1242.3; 1363.3; 1466.1; 1562.3; 2935.8; 3216.6; ^1^H NMR (600 MHz, DMSO-*d*_6_) (δ ppm.): 13.89 (1H, s, HCl), 8.82 (1H, s, Ar), 8.47 (1H, d, *J* = 8.5 Hz, Ar), 8.22 (1H, d, *J* = 5.8 Hz, Ar), 7.87 (1H, t, *J* = 7.8 Hz, Ar), 7.81 (1H, d, *J* = 8.9 Hz, Ar), 7.63–7.66 (2H, m, Ar), 3.72 (2H, q, *J* = 6.8 Hz, CH_2_), 3.24–3.30 (4H, m, CH_2_), 3.15 (2H, t, *J* = 7.8 Hz, CH_2_), 2.18 (2H, p, *J* = 7.9 Hz, CH_2_), 1.68 (2H, p, *J* = 8.1, 7.4 Hz, CH_2_), 1.54 (2H, p, *J* = 7.3 Hz, CH_2_), 1.31–1.42 (6H, m, CH_2_), (protons of NH groups invisible); MS (ESI) (M+1) *m/z*: 437.2, 185.1, 140.0; MS-HR (ESI) calcd. for C_25_H_30_Cl_2_N_4_O: 436.20189, found: 436.20299.

#### 3.2.7. 6-chloro-*N*-[3-(2,3-dihydro-1*H*-cyclopenta[*b*]quinolin-9-ylamino)octyl]pyridine-3-carboxamide hydrochloride (**3g**)

Yield: 48%; brown solid; mp 87–90 °C; IR (ATR)*v* (cm^−1^): 762.2; 1274.3; 1354.7; 1458.1; 1558.1; 2928.5; 3217.7; ^1^H NMR (600 MHz, DMSO-*d*_6_) (δ ppm.): 14.03 (1H, s, HCl), 8.83 (1H, s, Ar), 8.50 (1H, d, *J* = 8.8 Hz, Ar), 8.24 (1H, d, *J* = 8.3 Hz, Ar), 7.85–7.89 (1H, m, Ar), 7.84 (1H, d, *J* = 8.5 Hz, Ar), 7.62–7.65 (2H, m, Ar), 3.71 (2H, q, *J* = 6.7 Hz, CH_2_), 3.24–3.30 (4H, m, CH_2_), 3.15 (2H, t, *J* = 7.9 Hz, CH_2_), 2.16–2.20 (2H, m, CH_2_), 1.65–1.71 (2H, m, CH_2_), 1.50–1.57 (2H, m, CH_2_), 1.24–1.41 (8H, m, CH_2_), (protons of NH groups invisible); MS (ESI) (M+1)*m/z*: 451.2, 295.2, 185.1, 140.0; MS-HR (ESI) calcd. for C_26_H_32_Cl_2_N_4_O: 450.21864, found: 450.21936.

#### 3.2.8. 6-chloro-*N*-[3-(2,3-dihydro-1*H*-cyclopenta[*b*]quinolin-9-ylamino)nonyl]pyridine-3-carboxamide hydrochloride (**3h**)

Yield: 45%; brown solid; mp 95–97 °C; IR (ATR)*v* (cm^−1^): 760.9; 1273.9; 1399.5; 1417.7; 1558.2; 2922.4; 3204.7; ^1^H NMR (600 MHz, DMSO-*d*_6_) (δ ppm.): 14.07 (1H, s, HCl), 8.83 (1H, s, Ar), 8.50 (1H, d, *J* = 8.7 Hz, Ar), 8.24 (1H, d, *J* = 8.3 Hz, Ar), 7.84–7.89 (2H, m, Ar), 7.62–7.66 (2H, m, Ar), 3.71 (2H, q, *J* = 7.1 Hz, CH_2_), 3.24–3.31 (4H, m, CH_2_), 3.15 (2H, t, *J* = 7.9 Hz, CH_2_), 2.18 (2H, p, *J* = 7.9 Hz, CH_2_), 1.67 (2H, p, *J* = 7.1 Hz, CH_2_), 1.50-1.56 (2H, m, CH_2_), 1.22–1.41 (10H, m, CH_2_), (protons of NH groups invisible); MS (ESI) (M+1) *m/z*: 465.2, 309.2, 185.1, 140.0; MS-HR (ESI) calcd. for C_27_H_34_Cl_2_N_4_O: 464.23429, found: 464.23565.

### 3.3. Biochemical Evaluation

#### 3.3.1. In Vitro Inhibition Studies on AChE and BuChE

The inhibitory activities against EeAChE and EqBuChE of the test compounds were carried out based on the modified Ellman’s method [12,13]. Nine concentrations of each compound prepared in phosphate buffer (pH 8.0) were used to measure the inhibitory activity. The assay medium consisted of 0.4 mg/mL of 5,5’-dithio-bis(2-nitrobenzoic acid) (DNTB), 2 U/mL AChE or 4 U/mL BuChE, and ATCh (1 mM or 2mM for AChE and BuChE, respectively). Each concentration was analyzed in triplicate for 20 min at 30 °C after adding the substrate. The percent inhibition due to the presence of test compound was calculated. The concentration of the test compounds causing 50% inhibition (IC_50_) was calculated from the concentration-inhibition response curve.

#### 3.3.2. Kinetic Characterization of AChE Inhibition

Kinetic characterization of AChE and BuChE was performed using the reported method with different concentrations of the substrate (S)—acetylthiocholine iodide (in range of 0.05–0.50 µM) prepared in phosphate buffer (pH 8.0). The assay medium consisted of 0.4 mg/mL DTNB, 2 U/mL AChE and 40 µL of substrate. Then, the inhibitor (14 µL) was added into the assay solution. Kinetic characterizations of the hydrolysis of acetylthiocholine chloride catalyzed by AChE were done spectrophotometrically at 412 nm. A parallel control with no inhibitor in the mixture, allowed adjusting activities to be measured at various times.

#### 3.3.3. Beta-Amyloid Assay

Amyloid β aggregation plays a definitive role in neurodegeneration and AD progression. Thus, inhibition of the Aβ aggregation has been presented as a promising approach for the AD therapy [24,25,26]. In order to examine the ability to inhibit Aβ aggregation spectroscopic technique which is analyse with thioflavin T have been used. We investigated the best potent inhibitor towards AChE. Synthesized compound (**3e**) was used in four different concentrations (10, 25, 50, 100 µM). As well as the study being conducted without the presence of the inhibitor. Aβ_42_ was dissolved in DMSO to make 125.8 µM stock solution. Each compound (10 µL) and 10 µL of Aβ_42_ were added into 60 µL of phosphate buffer (PBS at pH 8.0) in a 96-well plate. After shake and incubation for 24 h at 30 °C, 20 µL of 2.5 µM ThT solution in 50 mM glycine–NaOH (pH 8.5) was added. Fluorescence excitation was measured at 446 nm with emission a wavelength of 490 nm after 5 min incubation with the dye. The reported results were obtained as the mean ± SD of triplicates of three independent experiments. Data are reported as mean ± SD of at least 3 independent experiments.

#### 3.3.4. Cell Culture and Cytotoxicity Assay

ATCC CCL-110 cell line (normal, human fibroblasts derived from skin) was used in the experiment. Cells were cultured in EMEM medium (ATCC) and seeded in 96-wells plate for 24 h. Incubation with tested compound at the chosen concentrations: 0.109, 0.218, 0.437, 0.655, 0.873 µM and 1.091 µM (compound was dissolved in methanol) lasted 48 h. Final methanol concentration in each well was 1 µL per 100 µL of medium. EMEM medium with the addition of 1 µL methanol was used as a control and a positive control was pure methanol. After 48 h of incubation, the cytotoxicity test was carried out according to Vybrant MTT Cell Proliferation Assay Kit: 10 µL of the 12 mM MTT stock solution was added to the cells and further incubated at 37 °C for 4 h. After incubation, not all medium but 25 µL was removed from the wells and 50 µL of DMSO was added. Plates were incubated for 10 min at 37 °C and absorbance at 540 nm was read [27,28,29].

#### 3.3.5. Cell Culture and Determination of Hepatotoxicity

Human hepatic stellate cells (HSCs, Sciencell, Carlsbad, CA, USA) were cultured in the incubator (37 °C, 5% CO_2_) in the Stellate Cell Medium (Sciencell) supplemented with 2% of fetal bovine serum (Sciencell), 1% of Stellate Cell Growth Supplement (Sciencell) and 1% of penicillin/streptomycin solution (Sciencell). To conduct experiment, cells were seeded into 96-well plates at density of 5 × 10^3^ cells per well. They were cultured for 24 h in the incubator at 37 °C and 5% CO_2_. After incubation, medium was removed, and cells were exposed to the 100 µL of the compound solutions over a range of concentrations (10–0.1 µM) or nothing but culture medium (blank control). Cells were incubated for 24 h with the compound solutions. Then medium was removed and 50 µL of the MTT solution was added to each well and incubated in the dark for the next 2 h at 37 °C. After incubation, the MTT solution was removed and 100 µL of DMSO was added. Plates were incubated for 10 min at room temperature. Then, 5 µL of Sorensen buffer was added to each well. Plates were swayed, and the absorbance was measured at a wavelength of 570 nm by microplate reader (Synergy H1, BioTek, Winooski, VT, USA). The cell viability was expressed as a percentage of the control values (blank) [30,31,32].

#### 3.3.6. Hyaluronidase Inhibition Test

The inhibition study of hyaluronidase was set by a turbidimetric method described previously [13,33]. The assay started by adding 20 µL of tested compound in monosodium phosphate buffer and then 40 µL of hyaluronidase solution (22.5 U/mL, Sigma Aldrich, Poznan, Poland) to the wells of 96-wells plates. The mixture was kept in the dark for 10 min at the temperature 37 °C. After incubation, 40 µL of hyaluronic acid solution (0.03%, Sigma Aldrich) in monosodium phosphate buffer was added to the wells. The mixture was kept in the dark for 45 min at the temperature of 37 °C. Finally, 300 µL of bovine serum albumin (0.1%, Serva) in sodium acetate buffer was added to the mixture and incubated for 10 min at room temperature. Changes in turbidity were measured by a microplate reader (BioTek, Winooski, VT, USA) at 600 nm. Heparin (WZF, Polfa, Warsaw, Poland) was a positive control. The assay was carried out in triplicate. The inhibitory activity of the tested compound was calculated from the equation [13,33]: (1)$$\%\mathrm{inhibition}=100x\left(1-\left(\frac{A_{HA}-A_{AN}}{A_{HA}-A_{HYAL}}\right)\right)$$where *A_HA_*—absorbance of solution without the enzyme (positive control), *A_HYAL_*—absorbance of solution without the tested compound (negative control), *A_AN_*—absorbance of solution with the tested compound.

#### 3.3.7. pKa Assay

Potassium dihydrogen phosphate, potassium hydroxide, methanol (POCH) were used to prepare buffer solution. Stock phosphate buffer was prepared from 500 mL 0.02 M potassium dihydrogen phosphate solution by adding 500 mL methanol. Thirty five work buffers were prepared by titrating stock phosphate buffer by 0.1 M potassium hydroxide solution in methanol:water (1:1) mix. The measurement of pH was performed at 23 °C by using pH-meter Mettler Toledo FiveEasy with Lab pH electrode LE438 (Mettler Toledo, Greifensee, Switzerland). Each work buffer was performed by collecting 25 mL of titrated stock phosphate buffer starting at pH 5.6, then every 0.2 pH value to pH 12.4. Our tested compound solution was 5 µM solution in methanol:water (1:1) mix.

Spectrophotometric measurement was performed in 96 wells plate by using Synergy H1 microplate reader (BioTek, Vinoosky, VT, USA) with Gen5 software (BioTek). The full assay was consisted of 35 UV spectra measurements, one for each work buffer solution. For assay, in each 35 wells was added 180 µL sequent work buffer and 20 µL tested compound solution. For blank, in each 35 well was added 200 µL sequent work buffer only. The measurement was performed at 23 °C. The spectral range was set from 280 nm to 380 nm by 1 nm step. Obtained spectra were subtracted from blank. Further calculations on UV spectra were performed by using mathematical ratios of absorbance 332/343 nm and 343/332 nm and according to methodology developed by Musil et al. [34]. Specific pH value for every ionized form of molecule was estimated by finding the smallest difference between fallowing values of absorbance ratios. Average of pKa values calculated for both ratios was considered as the final value of pKa. Experimental results were compared with computer calculated values. Computer calculations of the pKa value were performed by online software chemicalize.com (ChemAxon 2018) and ACD/Percepta version 14.0.0 (Advanced Chemistry Development, Inc., Metropolitan Toronto, ON, Canada).

#### 3.3.8. LogP Assay

Methanol (POCH) was used as an organic modifier. Demineralized water was purified in our faculty. Triethylamine (TEA) (Sigma Aldrich) was dissolved in concentration 30 mM in methanol and in water. Ten various isocratic mobile phases were used in calibration and assay. The first mobile phase contained 50% methanol solution with 30 mM TEA and 50% water solution with 30 mM TEA. Each next mobile phase contained 5% more methanol solution; 95% methanol solution and 5% water solution mix was used as the last mobile phase; 30 mM triethylamine in all mobile phases allowed us to analyzed basic substances. Basic solutions were used, because a neutral form of the compounds was required in logP assay. The basic properties of the mobile phase did not affect the retention times of neutral compounds.

Similarity in structure to the test compounds was the most important properties in selection of calibration compounds. Stock solutions of calibration compounds and test compound contained about 1 mg/mL. The injection concentration was 100 ug/mL.

The Waters 600 HPLC System composed of the 996 photodiode array detector, 600 solvent pump, 600 controller, 717 autosampler and PC with Waters Millennium software was used. Detector was set at respective optimum absorption wavelength for each compound. The chromatographic column Waters Xbridge C18 50 mm × 4.6 mm i.d., 3.5 µm was used. 

A modified procedure originally developed by Chao Liang et al. was used in this assay [35]. All calibration substances were eluted by all mobile phases to achieve their retention times. All retention times obtained were placed in the equation.

(2)$$\log k=\log\left(\frac{tR-t_{0}}{t_{0}}\right)$$

Value *t_0_* was a dead time. Sets of logk values were used to linear regression to calculate *logk_w_*. Effect of all this procedure were calibration curve (Figure 3).

The test compound was eluted by all mobile phases and obtained retention times were used to calculate *logk_w_* value. LogP value was read from calibration curve. Obtained value was compared with calculated values by online software chemicalize.com (ChemAxon 2018) and ACD/Percepta Version 14.0.0 (Advanced Chemistry Development, Inc., Metropolitan Toronto, ON, Canada).

#### 3.3.9. A Yeast Three-Hybrid (Y3H) Technology Test

The Y3H method was applied to evaluate if the synthesized hybrid ligand induces in vivo the interactions between AChE and A4, BACE1A, MAO B and MAPT. Plasmids suitable for Y3H screening were prepared by Gene Universal Inc. (Newark, DE, USA). The company performed cDNA synthesis, molecular cloning of inserts in a proper frame as the NdeI/BamHI fragments into the pGBKT7 (ACHE) or pGADT7 (A4, BACE1A, MAO B, MAPT) vectors. Gene Universal Inc. performed also the yeast codon optimization to facilitate the high expression of human recombinant proteins in *Saccharomyces cerevisiae*. From the cDNA encoding for human AChE (GenBank M55040.1) fragments were removed for signal peptide (aa 1–51) and the domain responsible for protein tetramerisation (aa 578–611). Therefore, to prepare the pGBKT7-hAChE plasmid the cDNA fragment encoding for aa 52–577 was used. The next plasmid pGADT7-hBACE-1A was built from the extracellular, N-terminal domain (aa 46–457) of BACE-1A known also as BACE-501, containing the enzyme active site (GenBank: AF204943.1). As a result, fragments encoding for signal peptide (aa 1–21), propeptide (aa 22–45), transmembrane (aa 458–478) and cytoplasmic (aa 479–501) domains were removed from the hBACE1A-pGADT7 construct. The pGADT7-hA4 plasmid was prepared from the human amyloid A4 protein fragment (UniProtKB-P05067-1), encoding for the 42 aa (aa 672–713). The pGADT7-hMAPT plasmid was obtained from the cDNA encoding for the entire protein (aa 1–758) (UniProtKB-P10636-1). Finally, the pGADT7-hMAO B construct was prepared from the cDNA encoding for the cytoplasmic domain (aa 1–489) of human MAO B. Prepared plasmids were used to transform competent *S. cerevisiae* cells according to the manual of the Matchmaker Gold yeast two-hybrid system (Takara/Clontech, Mountain View, CA, USA). The pGBKT7-hACHE plasmid was used to transform the Y2H Gold strain. Moreover, the remaining four plasmids were applied to transform Y187 strain. Transformed yeast strain Y2HGold [pGBKT7-hACHE] was tested for bait autoactivation. All negative and positive controls described in the manual of Y2H Matchmaker Gold system were performed.

The small scale mating procedure (5mL) based on Matchmaker Gold Y2H system manual recommendations was applied to exclude the putative protein–protein interactions between bait (AChE) and four preys (A4, BACE1A, MAO B, MAPT). Obtained cell suspension was transferred on DDO agar plates to initially screen for putative protein–protein interactions. DDO agar plates were supplemented with Aureobasidin A (200 ng/mL) and X-α-Gal (40 µg/mL) (DDO/X/A).

Finally, a small scale mating procedure (10 mL) was performed. The mating mixture contained the hybrid ligands at the concentration of 10 µM to promote the putative ligand-mediated protein interactions. The products of mating were placed on DDO/X/A agar plates containing 10 µM of hybrid ligand. The blue colony was transferred to the more stringent QDO agar plates containing Aureobasidin A (200 ng/mL), X-α-Gal (40 µg/mL) (QDO/X/A) and hybrid ligand (10 µM) to confirm the presence of interactions. The obtained blue colony was spread on QDOX/A agar plates without the hybrid ligand to confirm that the interaction depends on the hybrid ligand. The strength of hybrid-ligand induced interactions was quantitatively evaluated by yeast β-galactosidase assay kit (ThermoFisher Scientific, Waltham, MA, USA).

#### 3.3.10. ADMET Analysis

ADMET (Absorption, Distribution, Metabolism, Excretion, Toxicity) prediction was performed to estimate the risk and effectiveness of use our compound as a medicine. Calculated parameters were confronted with the values defined by the Lipinski’s “Rule of Five”. Our experimental parameters like logP and pKa (Base) were used as basic parameters to predictions. The ADMET analysis was done with the help of ACD/Percepta Version 14.0.0 (Advanced Chemistry Development, Inc., Metropolitan Toronto, ON, Canada).

#### 3.3.11. Antioxidant Activity Assay

DPPH Assay. The antioxidant activities of compounds **3a**–**3h** were evaluated by 1, 1-diphenyl-2-picrylhydrazyl (DPPH) free-radical scavenging assay according to the method of Blois (1958) with slight modifications [36]. Briefly, 50 µL of DPPH radical solution (0.2 mM) was added in a 96-well plate containing 50 µL of concentrations of test compound ranging from 0.15 to 150 mM dissolved in MeOH, and incubated for 30 min at room temperature in the dark. The absorbance of each well was measured at 517 nm using a microplate reader Synergy H1 (Biotek). The FRS_50_ values of test compounds were determined. Results are expressed as the mean ± SD of at least three different experiments performed in triplicate.

ABTS^•+^ Assay. The ABTS^•+^ solution was prepared by reaction of 5 mL of a 7 mM aqueous ABTS solution with 88 µL of a 140 mM potassium persulfate. After storage in the dark for 16 h, the solution was diluted until the absorbance value of 0.70 ± 0.05 at 734 nm was reached. Solutions of each tested compounds were prepared in PBS (pH = 7.4). For each compound and concentration, measurements were made in triplicate with blank solutions using a microplate reader Synergy H1 (Biotek) [37].

#### 3.3.12. Neuroprotection Against Oxidative Stress

##### Cell Culture

The SH-SY5Y (human neuroblastoma) (European Collection of Cell Culture) was chosen to determine the potential neuroprotective properties of novel compounds. SH-SY5Y were grown in Ham’s F12:EMEM (1:1) (Sigma Aldrich) containing 15% Foetal Bovine Serum (Biowest, Riverside, MO, USA), 2mM Glutamine (Sigma Aldrich), 100 units/mL penicillin and 100 mg/mL streptomycin (Biological Industries) and 1% non-essential amino acids (Biological Industries). Cells were grown in an incubator at 37 °C with 5% CO_2_ before the initiation of the assay.

##### MTT Assay

Cells were seeded in 96-well plates at the density of 5 × 10^3^ cells per well and cultured for 24 h in the incubator (37 °C, 5% CO_2_). After 24 h of cells incubation, medium was removed and cells were exposed to the 100 µL of the compound solutions over a range of concentrations ranging from 0.0001 to 10 µM or nothing but culture medium (control). After 24 h, the medium was removed and 50 µL of the MTT solution was added. Plates were incubated for additional 2 h in the incubator (37 °C, 5% CO_2_). Then, the MTT solution was removed and 100 µL of DMSO was added. Plates were incubated for 10 min at room temperature and 5 µL of Sorensen Buffer was added. The plate was swayed and the absorbance was measured in the microplate reader (Synergy H1, BioTek, Winooski, VT, USA) at a wavelength of 570 nm. The cell viability was expressed as a percentage of the control values (blank). The experiments were done in triplicate [30].

The neuroprotective properties of **3e** against oxidative stress were determined in three different experiments. Hydrogen peroxide (H_2_O_2_) was used to generate exogenous free radicals in the first experiment. SH-SY5Y were incubated with compounds in the range of concentrations (from 10 µM to 0.01 µM) for 24 h before the addition of H_2_O_2_ (100 µM). Then, H_2_O_2_ was added and the incubation in the presence of compounds as conducted for the next 24 h. In the second and third experiments, mix of rotenone (30 µM) with oligomycin A (10 µM) (R/O) was used to cause mitochondrial reactive oxygen species. A second experiment involved “pre-incubation”. SH-SY5Y were incubated with **3e** before the addition of the rotenone and oligomycin A mixture for 24 h. Next, the mixture was added, and cells were incubated with **3e** for additional 24 h. The last experiment, “co-incubation”, involved adding the cells at the same time as the mixture of rotenone with oligomycin A and **3e**. Incubation took 24 h. Trolox was used as a positive control. Each experiment was done three times in quadruplicates and cell death was tested by the MTT assay. Data were shown as the percentage of the reduction of MTT in regard to non-incubated cells [22,23].

#### 3.3.13. Molecular Modeling

The 3-dimentional structure of synthesized compounds were drawn in Corina on-line (Molecular Networks and Altamira) [38] and subsequently prepared using Sybyl 8.0 (Tripos, Certara, Princeton, NJ, USA). [39] Atom types were checked, hydrogen atoms were added and Gasteiger–Marsili charges were assigned. Acetylcholinesterase from the 2CKM and butyrylcholinesterase from the 1P0I crystal structures were prepared before docking in the following way: histidine residues were protonated at Nε, the hydrogen atoms were added, water molecules and ligands were removed. Docking was performed with GoldSuite 5.1 (CCDC). [40] We defined binding sites as all amino acid residues within a radius of 10 Å from bis-(7)-tacrine for AChE and 20Å from the glycerol molecule for BuChE. The standard settings of the genetic algorithm with a population size 100, number of operations 100,000 and clustering tolerance of 1 Å was applied. As a result, we obtained 10 ligand poses, sorted by GoldScore (for AChE) and ChemScore (for BuChE) function value. To visualize the results, we used PyMol 0.99rc6 (DeLano Scientific LLC, Schrodinger, Cambridge, MA, USA) [41].

## 4. Conclusions

To sum up, a novel series of eight cyclopentaquinoline derivatives were designed, synthesized, and evaluated as potential multifunctional compounds for the AD therapy. In vitro studies concerned their ability to inhibit AChE and BuChE. The results demonstrated that compound **3e** is a dual AChE and BuChE inhibitor and a promising compound for further development. This compound is a good EeAChEI and EqBuChEI with IC_50_values (0.067 and 0.153 µM, respectively). **3e** compound shows a mixed-type of inhibition confirmed by in vitro study and molecular modeling. What is very important is that **3e** derivative are non-hepatotoxic, show low antioxidant activity, and have the ability to inhibit Aβ aggregation. The results of ADMET prediction and LogP assay, which confirmed the fulfillment of Lipinsky’s rule of five, shows a good pharmacokinetics and future potential use in medicine. Moreover, we conducted an innovative study—a yeast three-hybrid technology (Y3H) which is an extension of the two-hybrid system (Y2H). In the Y2H, the interaction between two hybrid proteins activates the expression of reporter genes facilitating the yeast cells to grow on particular selective media [42,43]. In the Y3H method, the interaction between two hybrid proteins is mediated by a hybrid ligand molecule. Therefore, the Y3H technology may be used to confirm the known ligand–receptor interactions or to identify new small molecules interacting with known receptors [44,45,46,47]. In the presented paper, the Y3H method was used to evaluate in vivo the hybrid ligand-mediated interactions between human acetylcholinesterase (hAChE) and four proteins: human amyloid beta A4 (A4), human beta-secretase 1A (BACE-1A), human monoamine oxidase B (MAO B) and human microtubule-associated protein tau (MAPT) and the results suggest interaction only between AChE and BACE1A. Hybrid ligands promoted interaction between these two proteins that do not interact on their own.

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
