# Peer review of "Discovery of New Cyclopentaquinoline Analogues as Multifunctional Agents for the Treatment of Alzheimer’s Disease"

_ijms, 2019, doi:10.3390/ijms20030498_

Round 1
Reviewer 1 Report
The manuscript by Czarnecka et al. describes the synthesis and pharmacological evaluation of a set of 8 hybrids of 2,3-dihydro-1H-cyclopenta[b]quinoline-9-amine (i.e. a ring contraction analogue of tacrine that has been previously exploited by the same group, see below) and 6-chloronicotinic acid, using tether chains from 2 to 9 methylenes. The Authors have previously reported a parallel set of compounds with the same cyclopentaquinolinamine moiety and the same linkers but with 5,6-dichloronicotinic acid instead of 6-chloronicotinic acid (mentioned in the text (ref. 13) as well as compounds with the same cyclopentaquinolinamine moiety and a 6-hidrazinonicotinic acid. It is, hence, obvious that the new compounds described in this manuscript do not have too much structural novelty. The same applies to the tested biological activities and observed potencies.
The best compounds display in vitro nanomolar potencies against electric eel acetylcholinesterase (AChE) and equine serum butyrylcholinesterase (BuChE), which are similar to or lower than the AChE and BuChE inhibitory potencies of the known anticholinesterase anti-Alzheimer drug tacrine. The inhibitory activity of one selected compound against the in vitro aggregation of the beta-amyloid peptide was assessed, to find a moderately potent micromolar activity. This pattern of potencies is very common in many tacrine derivatives.
In vitro cytotoxicity assays with that selected compound revealed a submicromolar IC50 value for toxicity in human fibroblast CCL-110 cells (no comparison with tacrine is given) and similar toxicities to tacrine (considered a hepatotoxic drug) in human hepatic stellate cells.
The Authors have performed other assays, whose rationale is not explained or does not make much sense, which do not add any value to the work: hyaluronidase inhibition (the selected compound is very weak, IC50 651 micromolar), pKa assay (tacrine derivatives are well-known to be basic compounds, with pKa values around 8-9), logP (it can be readily calculated), yeast three-hybrid test (that suggests that the selected compound induces an interaction between AChE and BACE-1 ??? which is the relevance of this result?), antioxidant activity (the compounds are either inactive at all or have IC50 values in the millimolar range !!!, so not active too), neuroprotection assays against oxidative stress (these assays do not make any sense as the compounds were found inactive as antioxidants; quite expectedly no neuroprotective activity was found in SH-SY5Y cells using different oxidative insults).
In summary, this manuscript reports on a set of compounds which are quite similar to other previously reported by the group, whose interest lies in their anticholinesterase activities, which are in the range of many other tacrine derivatives.
I have an additional concern about the suitability of the 6-chloronicotinic acid moiety that is present in the hybrids reported in this work, because it is highly electrophilic and could arylate nucleophilic endogenous macromolecules leading to toxicity. Indeed, this moiety raises a Brenk alert of toxicity or unstability (see SwissADME, for example) and could be responsible for the cytotoxicity observed for these compounds.
The Authors describe very honestly the results of this work, but, in my opinion, neither the target compounds nor the obtained results are of much interest in the field of multitarget anti-Alzheimer agents. I recommend publication in another lower impact factor journal.
To that purpose, I recommend some points to be addressed by the Authors:
1) The manuscript needs in depth English editing.
2) The chemical name given in the abstract should be written with proper use of italics and blank spaces.
3) Figure 1 is absolutely superfluous, it should be removed.
4) The chemical characterization of the compounds should include 13C NMR spectra. In section 4.1. it is said that elemental analyses were done but they are not included in the chemical characterization of the compounds.
5) I would suggest to removing all parts of the manuscript that describe the assays that do not have a clear rational basis and do not afford value (see above), just a few sentences stating the lack of activity in these assays would be more than enough.
6) Section 4.4.6., the calculated and found values given for the HRMS should be interchanged and one of them is wrong.
7) Line 542, acetylthiocholine instead of acetylcholine.
8) The journal names in some references are not written in the abbreviated form.
Author Response
Reviewer 1
Thank you so much for your minute observation and valuable comments.
Comment:
The manuscript by Czarnecka et al. describes the synthesis and pharmacological evaluation of a set of 8 hybrids of 2,3-dihydro-1H-cyclopenta[b]quinoline-9-amine (i.e. a ring contraction analogue of tacrine that has been previously exploited by the same group, see below) and 6-chloronicotinic acid, using tether chains from 2 to 9 methylenes. The Authors have previously reported a parallel set of compounds with the same cyclopentaquinolinamine moiety and the same linkers but with 5,6-dichloronicotinic acid instead of 6-chloronicotinic acid (mentioned in the text (ref. 13) as well as compounds with the same cyclopentaquinolinamine moiety and a 6-hidrazinonicotinic acid. It is, hence, obvious that the new compounds described in this manuscript do not have too much structural novelty. The same applies to the tested biological activities and observed potencies.
Response:
Thank you so much for your suggestion. ADMET analysis and proviusly described results give us information than we are at good direction to obtain the active structure in AD therapy. That is why we decided to to slightly modify the structure of previously tested compounds and examine them more thoroughly than previously published compounds.
Comment:
The best compounds display in vitro nanomolar potencies against electric eel acetylcholinesterase (AChE) and equine serum butyrylcholinesterase (BuChE), which are similar to or lower than the AChE and BuChE inhibitory potencies of the known anticholinesterase anti-Alzheimer drug tacrine. The inhibitory activity of one selected compound against the in vitro aggregation of the beta-amyloid peptide was assessed, to find a moderately potent micromolar activity. This pattern of potencies is very common in many tacrine derivatives.
Response:
Thank you so much for your comment. This test is performed commonly in similar concentrations by many research teams. To compare the results we decided to use these concentrations. Obtained values are satisfying in the area of searching the drugs in AD therapy. What is more compounds have other very promising properties.
Comment:
In vitro cytotoxicity assays with that selected compound revealed a submicromolar IC50 value for toxicity in human fibroblast CCL-110 cells (no comparison with tacrine is given) and similar toxicities to tacrine (considered a hepatotoxic drug) in human hepatic stellate cells.
Response:
Thank you so much for your comments. Tacrine at very low, one-administrated concentrations is not toxic, what was confirmed in our publication and what is confirmed by other authors. Toxicity appears during longer use. In vitro assay on human hepatic stellate cells is a preliminary test, which is performed by many researchers worldwide (exactly this kind of test) and which appears in many publications regarding Alzheimer’s disease. In this preliminary test we checked if novel compound has lower or higher toxicity in comparison to tacrine after 24 hours of incubation. What was interesting, in higher concentration (10 µM), novel compound was less toxic than tacrine, which is a very good result. What is also crucial, authors used normal hepatic cells, not cancer cells. To confirm real hepatotoxicity effect on liver, in vivo assays such as “repeated dose 28-day oral toxicity study in rodents” in regard to OECD 407 will be performed, for which authors obtained already a consent from the Local Ethics Committee. This test could really confirm whether novel compound has hepatotoxic effect. However, in vivo assays cannot be performed without results from in vitro assays. Therefore, this in vitro assay on human hepatic stellate cells was performed and will be basis for next research.
For human fibroblast cells tacrine was not tested, because tacrine is not used on skin. Novel compound could be considered in transdermal application of a drug, therefore first assay on human skill cells was performed.
Comment:
The Authors have performed other assays, whose rationale is not explained or does not make much sense, which do not add any value to the work: hyaluronidase inhibition (the selected compound is very weak, IC50 651 micromolar), pKa assay (tacrine derivatives are well-known to be basic compounds, with pKa values around 8-9), logP (it can be readily calculated), yeast three-hybrid test (that suggests that the selected compound induces an interaction between AChE and BACE-1 ??? which is the relevance of this result?), antioxidant activity (the compounds are either inactive at all or have IC50 values in the millimolar range !!!, so not active too), neuroprotection assays against oxidative stress (these assays do not make any sense as the compounds were found inactive as antioxidants; quite expectedly no neuroprotective activity was found in SH-SY5Y cells using different oxidative insults).
Response:
Thank you so much for your comments. That is right that, we missed a correct explanation of performing pKa and logP assay in our study. Our goal was to investigate new, active chemical compounds including study of activity and prediction of pharmacokinetics. According to yours notes, we agree, that logP and pKa can be calculated, but we care about reliable results, therefore we performed physicochemical assays. Determination of pKa was an additional, auxiliary research necessary to performed logP assay correctly. We added comparison of experimental and estimated values of pKa and logP, because we feel it is important to mention the differences between them. These properties were used in ADMET prediction by ACD/Percepta version 14.0.0 software, because this program allows to enter experimental physicochemical data. Thanks to experimental data, we have the most reliable ADMET results as possible.
Comment:
Authors explained the rationale of assays and these assays make sense in regard to investigation of novel drug against Alzheimer’s disease: “The ‘oxidative stress hypothesis’ of AD, of which reactive oxygen species (ROS) plays a key role in AD onset and progression is well known. The relationship between oxidative stress and neurodegeneration is complex. Production of aberrant amounts of hydrogen peroxide is a general feature of aging and is linked with neurodegeneration.”
Response:
All assays, on enzymes and cell lines (SH-SY5Y) are model assays against oxidative stress and inflammation, and are performed by research teams in order to investigate novel compounds against Alzheimer’s disease. Therefore, also authors carried out tests to determine whether compounds could stop neurodegeneration. First, tests were performed on enzymes and later authors confirmed their results in in vitro assays on cell lines. Assays on enzymes and cell lines should be paralleled and should have the same relation, what was confirmed. Obtained results showed that compounds did not have antioxidant activities. But without these tests, the antioxidant properties would remain unknown.
Comment:
In summary, this manuscript reports on a set of compounds which are quite similar to other previously reported by the group, whose interest lies in their anticholinesterase activities, which are in the range of many other tacrine derivatives.
Response:
Thank you so much for your suggestion. In the search for a new substance effective in AD therapy, inhibition of cholinesterases is just one of many goals. Due to the complicated mechanism of disease formation, the tested compounds have been studied in many directions. We also used modern methods such as the yeast three-hybrid technology.
Comment:
I have an additional concern about the suitability of the 6-chloronicotinic acid moiety that is present in the hybrids reported in this work, because it is highly electrophilic and could arylate nucleophilic endogenous macromolecules leading to toxicity. Indeed, this moiety raises a Brenk alert of toxicity or unstability (see SwissADME, for example) and could be responsible for the cytotoxicity observed for these compounds.
Response:
Thank you so much for your minute observation which will be valuable in our further work. Electrophilic moiety makes it easier to pass the blood brain barrier and that's why we decided to connect it.
Comment:
1) The manuscript needs in depth English editing.
Response: Thank you so much for your comment. The manuscript was edited.
2) The chemical name given in the abstract should be written with proper use of italics and blank spaces.
Response: Thank you so much for your comment. The names were corrected.
3) Figure 1 is absolutely superfluous, it should be removed.
Response: Thank you so much for your comment. Figure 1 was removed.
4) The chemical characterization of the compounds should include 13C NMR spectra. In section 4.1. it is said that elemental analyses were done but they are not included in the chemical characterization of the compounds.
Response: It was our mistake, we mentioned Elemental analysis in Chemistry but in fact we didn't it. It was corrected. All target compounds were characterized by 1 H NMR, ESI-MS, MS-HR and IR. Instead 13CNMR we prepared both type of MS analysis: Hight Resolution MS and full mass spectrum for the characterization of fragments and whole structure.
5) I would suggest to removing all parts of the manuscript that describe the assays that do not have a clear rational basis and do not afford value (see above), just a few sentences stating the lack of activity in these assays would be more than enough.
Response: For us the manuscript consists of describing results, not describing only what gives positive effects.
6) Section 4.4.6., the calculated and found values given for the HRMS should be interchanged and one of them is wrong.
Response: Thank you so much for your valuable suggestion. We have corrected it in the revised manuscript.
7) Line 542, acetylthiocholine instead of acetylcholine.
Answer: Thank you so much for your valuable suggestion. We have corrected it in the revised manuscript.
8) The journal names in some references are not written in the abbreviated form.
Response:
Thank you so much for your valuable suggestion. We have corrected it in the revised manuscript.

Reviewer 2 Report
The work presented here is an interesting study aimed to find possible AD therapeutic compounds. It is very well conducted and documented, however the reviewer has some questions.
Has the blood-barrier-barrier permeability measure measured? If not is should be interesting to measure it at least in silico. Moreoever, the CNS permeability measure could be also an interesting observable to evaluate the therapeutic capacity of these compounds.
Only rigid docking studies were performed? How many replica? The obtaining binding mode was reproduced in subsequent experiments? Probably an MD simulation to post-processing the obtained results will give a more realistic view of the binding mode. Just rigid docking results used to give poor information because they are performed over an static representation of a protein, that is biased to the cocrystal and thus several interactions could be missing and other that are present can dissapear after the observation of the induced fit events. So it is important to ensure the suggested binding mode.
Author Response
Reviewer 2
Thank you so much for your comments and suggestion.
Comment:
Has the blood-barrier-barrier permeability measure measured? If not is should be interesting to measure it at least in silico. Moreoever, the CNS permeability measure could be also an interesting observable to evaluate the therapeutic capacity of these compounds.
Response:
Thank you so much for your comment. ADMET prediction was performed for 3e with the best properties. We used experimental values of logP and pKa (base). 3epresent a good blood-brain barrier permeation, logPS value was equal -1.7. Compound can penetrate to brain tissue, logBB was equal 0.37 with fraction unbound in plasma 0.062 and fraction unbound in brain 0.03.
Comment:
Only rigid docking studies were performed? How many replica? The obtaining binding mode was reproduced in subsequent experiments? Probably an MD simulation to post-processing the obtained results will give a more realistic view of the binding mode. Just rigid docking results used to give poor information because they are performed over an static representation of a protein, that is biased to the cocrystal and thus several interactions could be missing and other that are present can dissapear after the observation of the induced fit events. So it is important to ensure the suggested binding mode.
Response:
Thank you so much for your comments. Each compound was docked ten times. The presented poses were the top-ranked and also the most frequent. With respect to fully felxible procedures, such as MD, it really allows to fit ligand into protein and protein to ligand. This is very important, especially in case of very flexible proteins. However, the structure of cholinesterases in complexes with inhibitors is relatively rigid. Only in case of acetylcholinesterase, two amino acid residues are flexible, i.e. Phe330 and Trp279. The crystal structures with PDB codes: 1EVE, 1ACJ and 2CKM represent three the most frequent combinations of conformations of these two amino acids. At the step of validation we selected 2CKM structure as the most appropriate for docking of hybrid compound – derivatives of tacrine. Comparing our previous experiments with rigid and flexible docking to AChE it is worth to note that the results were similar. However, we are planning to add molecular dynamics simulations to our workflow in case of cholinesterase.

Reviewer 3 Report
The manuscript presents very important results in the development of new drugs taking into account the experimental results and correlating them with mathematical approximations about their biological activities, especially for the therapy of Alzheimer's disease.
There are only formal suggestions for the improvement of the final text, which are highlighted in yellow in the pdf document.
It is important to review all the significant figures and the assignment of the signals in nmr, according to the numbering of the carbon atoms that must be placed in the structures of scheme 1.

Author Response
Reviewer 3
Thank you very much for your valuable suggestion.
Comment:
There are only formal suggestions for the improvement of the final text, which are highlighted in yellow in the pdf document.
It is important to review all the significant figures and the assignment of the signals in nmr, according to the numbering of the carbon atoms that must be placed in the structures of scheme 1.
Response:
Thank you so much for your minute observation and valuable comments. We have taken reviewer’s comments in full consideration and it will be well reflected by the revised version of manuscript.

Reviewer 4 Report
This is a good manuscript, original, showing new results, that have been obtained in very well designed experiments. The paper is well presented, in a fine English.To sum up, this manuscript represents a very good contribution to the identification of new agents for the potential treatment of Alzheimer's disease, and consequently, I suggest to accept it as it is for publication in this Journal.
Round 2
Reviewer 1 Report
In my opinion, the Authors have not made much effort to improve the work. The reported results are devoid of significant novelty and relevance, and this is a major flaw inherent to this work. The work includes some assays whose rationale is neither properly explained nor understandable, which, additionally do not add any value to the biological characterization of the compounds. One thing is hidding information about negative results (I have never suggested that) and another thing is making a lot of meaningless or even unjustified assays, which, in addition, have afforded very poor results, and include them as important sections in the manuscript but without explaining clearly why they have been done, etc. As I said in my first report, with very short statements on the results of these assays would be more than enough (hence, not hidding information while making the manuscript much easier to read, without distracting the reader attention to irrelevant sections).
Reviewer 2 Report
Thank you very much for your answers. All the points are clear for me now.